# Left Ventricular Diastolic and Systolic Functions in Patients with Hypothyroidism [note 1]

**DOI:** 10.3390/medicina56100524

**Published:** 2020-10-07

**Authors:** Rina Tafarshiku, Michael Y. Henein, Venera Berisha-Muharremi, Ibadete Bytyçi, Pranvera Ibrahimi, Afrim Poniku, Shpend Elezi, Gani Bajraktari

**Affiliations:** 1Clinic of Endocrinology, University Clinical Centre of Kosova, 10000 Prishtina, Kosovo; rinatafarshiku@gmail.com (R.T.); venera.berisha@uni-pr.edu (V.B.-M.); 2Medical Faculty, University of Prishtina, 10000 Prishtina, Kosovo; afrimponiku@hotmail.com (A.P.); selezi@hotmail.com (S.E.); 3Institute of Public Health and Clinical Medicine, Umeå University, SE-901 87 Umeå, Sweden; michael.henein@umu.se (M.Y.H.); i.bytyci@hotmail.com (I.B.); pranvera_i86@hotmail.com (P.I.); 4Clinic of Cardiology, University Clinical Centre of Kosova, 10000 Prishtina, Kosovo; 5UBT—University for Business and Technology, 10000 Prishtina, Kosovo

**Keywords:** hypothyroidism, left ventricular function, Doppler echocardiography

## Abstract

*Background and objectives*: Long standing hypothyroidism may impair myocardial relaxation, but its effect on systolic myocardial function is still controversial. The aim of this study was to investigate left ventricular (LV) systolic and diastolic function in patients with hypothyroidism. *Materials and Methods*: This study included 81 (age 42 ± 13 years, 92% female) patients with hypothyroidism, and 22 age and gender matched controls. All subjects underwent a detailed clinical examination followed by a complete biochemical blood analysis including thyroid function assessment and anthropometric parameters measurements. LV function was assessed by 2-dimensional, M-mode and Tissue-Doppler Doppler echocardiographic examination performed in the same day. *Results*: Patients had lower waist/hip ratio (*p*< 0.001), higher urea level (*p* = 0.002), and lower white blood cells (*p* = 0.011), compared with controls. All other clinical, biochemical, and anthropometric data did not differ between the two groups. Patients had impaired LV diastolic function (lower E wave [*p*< 0.001], higher A wave [*p* = 0.028], lower E/A ratio [*p*< 0.001], longer E wave deceleration time [*p* = 0.01], and higher E/e’ ratio [*p*< 0.001]), compared with controls. Although LV global systolic function did not differ between groups, LV longitudinal systolic function was compromised in patients (lateral mitral annular plane systolic excursion—MAPSE [*p* = 0.005], as were lateral and septal s’ [*p*< 0.001 for both]). *Conclusions*: In patients with hypothyroidism, in addition to compromised LV diastolic function, LV longitudinal systolic function is also impaired compared to healthy subjects of the same age and gender. These findings suggest significant subendocardial function impairment, reflecting potentially micro-circulation disease that requires optimum management.

## 1. Introduction

Hypothyroidism is an endocrine disorder, caused by thyroid hormone deficiency, which may affect the heart and the cardiovascular system. Thyroid hormones assist in maintaining low peripheral resistance in peripheral arterioles through a direct effect on vascular smooth muscles, thus increasing blood volume, systemic vascular resistance, preload, and cardiac output in patients with hyperthyroidism [1].

In contrast, hypothyroidism have an opposite effect on the cardiovascular system and may reduce cardiac output by 30–50% [2]. It has a direct effect on cardiac function through changes in myocyte—specific gene expression [2,3]. It does also affect the arterial system, even at the endothelial level, causing dysfunction and impaired vascular smooth muscle (VSM) relaxation, thus leading to increased systemic vascular resistance and diastolic hypertension, which has been reported in approximately 30% of such patients [4]. Despite the available evidence on the impairment of the cardiovascular system in patients with hypothyroidism, results remain inconsistent and could even be controversial [5,6,7,8,9,10,11].

The aim of this study was to investigate the effect of hypothyroidism on left ventricular (LV) systolic and diastolic function components using conventional Doppler echocardiography.

## 2. Patients and Methods

### 2.1. Patients

This study included 81 consecutive patients (age 42 ± 13 years, 92% female) diagnosed with clinical hypothyroidism at the Outpatient Endocrinology Clinic of the University Clinical Centre of Kosovo, and 22 subjects, with normal thyroid and renal function who served as a control group, recruited between May 2017 and March 2018. All subjects gave informed consent to participate in the study, which was approved by the Ethics Committee of Medical Faculty, University of Prishtina, with reference number 4056, date 20 June 2014. All included patients had been diagnosed with hypothyroidism depending on thyroid hormone levels and clinical examination by an experienced endocrinologist. Patients with decompensated heart failure, active malignancy, hepatic or pulmonary disease, pregnant women, and those with arterial hypertension were excluded.

### 2.2. Clinical Data

In all participants, demographic details, physical examination and anthropometric measurements were taken. Hypothyroidism was defined by low serum T3, T4 levels, high serum thyroid-stimulating hormone (TSH) level, and elevated circulating anti-peroxidase level. Body mass index (BMI) was measured and was calculated by dividing dry weight by body height (kg/m^2^). Blood pressure was recorded with a brachial sphygmomanometer, after the subject had rested in the supine position for at least 10 min.

### 2.3. Blood Analysis

Erythrocyte sedimentation rate, hematological investigations, fasting plasma glucose, blood urea nitrogen (BUN), creatinine, total cholesterol, and triglyceride were measured using standard methods. Thyroid hormones and anti-peroxidase were measured by Electrochemiluminescence (COBAS INTEGRA^®^ E 411 Roche). All samples for a given assay were tested simultaneously, in duplicate and in appropriate dilutions, according to conventional protocols.

### 2.4. Cardiac Structure and Function

LV structure measurements: Echocardiographic examinations were performed in all patients and controls at the same day of blood tests. LV volumes and ejection fraction (EF) were calculated from the apical 2- and 4-chamber views using the modified Simpson’s method. Left ventricular mass (LVM) was estimated using the anatomically validated formula of Devereux et al. [12,13] and was indexed to height ^2.7^ (LVMI) [14]:LVM(g) = 0.8 × (1.04 × (LVEDD + PWTD + IVSTD)^3^ − (LVEDD)^3^) + 0.6 [13].

Left ventricular hypertrophy (LVH) was defined as a LVMI of 47 g/m^2.7^ for women and 53 g/m^2.7^ for men [15]. LVM normalized for body surface area (LV mass/BSA) was also calculated as g/m^2^.

LV function measurements: From the spectral wave Doppler LV filling, peak E wave velocity, peak A wave velocity, the ratio between peak E and A velocities (E/A ratio), and E wave deceleration time were all measured. From tissue-Doppler imaging recordings, peak systolic (s’), early diastolic (e’), and late diastolic (a’) mitral annular velocities were also measured. The ratio of trans-mitral to myocardial early diastolic peak velocity (E/e’) was calculated, having averaged septal and lateral e’ velocities [16,17], to reflect filling pressures. Mitral annular plane systolic excursion (MAPSE) was measured by placing the M-mode cursor at the lateral and septal angles [18].

Total LV filling time (FT) was measured from the onset of the E wave to the end of the A wave and ejection time (ET) from the onset to the end of the aortic Doppler flow velocity. Total isovolumic time (t-IVT) was calculated as 60—(total ET + total FT) and was expressed in s/min [19]. Tei index was calculated as the ratio between t-IVT and ejection time [20].

Left atrial (LA) measurements: LA diameter was measured from aortic root recordings with the M-mode cursor positioned at the level of the aortic valve leaflets.

We followed the methods of our previous study, Poniku A et al. 2018 [21], which described in detail the above measurements.

### 2.5. Statistical Analysis 

Values are expressed as means ± standard deviation (SD). Differences between the two groups were analyzed using the unpaired Student *t* test following the analysis of variance. The Chi-square test was used to compare the categorical variables. *p* values <0.05 were considered statistically significant. All analyses were performed using SPSS 22 for Windows.

## 3. Results

Table 1 shows the baseline demographic, anthropometric, and clinical data of patients with hypothyroidism and controls. Age, gender, and CV risk factors (smoking, diabetes, mean blood pressure, and cholesterol level) were not different between groups. Patients with hypothyroidism had lower waist/hip ratio (*p* < 0.001), lower RBC (*p* = 0.019), and lower WBC (*p* = 0.011) compared to controls.

Patients with hypothyroidism did not differ from controls with regards to LV dimensions, global systolic function, stroke volume, LV wall thickness, LVMI, and LA dimensions. They, however, had lower lateral s’ and septal s’ (*p* < 0.001, for both), and lower lateral MAPSE (*p* = 0.002), compared to controls (Table 2).

Patients with hypothyroidism had higher A wave velocity (*p* = 0.028), lower E wave velocity, lower E/A ratio (*p* < 0.001, for both), lower E/e’ ratio, and longer E wave DT (*p* = 0.01), compared to controls (Table 2).

Patients with hypothyroidism did not differ from controls with regards to LV dyssynchroneous function parameters; isovolumic relaxation time, ejection time, filling time, total isovolumic time, and Tei index were not different between the two groups.

## 4. Discussion

### 4.1. Findings

Our results show that in a modest sample of patients with hypothyroidism and reduced waste/hip ratio, there was significant evidence for compromised LV longitudinal systolic function in the form of long axis amplitude of motion (MAPSE) and its systolic velocity. In addition, markers of LV diastolic function assessed by spectral Doppler were also quite abnormal with reduced E wave velocity, raised A wave velocity, and reduced E/A ratio, compared to controls. Other conventional measurements of LV structure and function including ejection fraction, stroke volume, LV mass, and synchronous function were not different when compared with controls.

### 4.2. Data Interpretation

Despite rather similar atherosclerosis risk factors, hypertension, diabetes, dyslipidemia and smoking, between groups, patients had significant evidence for impaired LV long axis systolic function which is known to determine, to a large extent, the overall cavity diastolic performance as shown by spectral Doppler in our patients. LV long axis function is subtended by the longitudinal myocardial fibers that constitute the subendocardial layer of the myocardium. This layer is known to be the most sensitive to ischemia, being vascularized by the distal coronary arterioles and capillaries [22]. Having excluded differences in conventional risk factors between groups, it seems that hypothyroidism remains the only pathology that might have impacted subendocardial function, in patients with hypothyroidism. It has previously been shown that hypothyroidism affects peripheral vascular resistance by effecting the vascular smooth muscle function [23]. Applying the same mechanism on the subendocardium could explain our findings. The other potential factor that could have played a role in impairing subendocardial function is overweight/obesity, manifested as reduced waste/hip ratio in our patients compared to controls. We and others have previously shown that obesity has a drastic effect on diastolic LV function even in young adults [24,25], through a number of biochemical disturbances and also the accumulated pericardial fat. Furthermore, many studies have already shown that diastolic dysfunction is hardly an isolated phenomenon, but likely reflect a degree of systolic dysfunction, even at subendocardial level [26,27]. Finally, it sounds plausible that the two factors could have played a role in causing the systolic and diastolic disturbances which we identified at the level of microcirculation.

### 4.3. Clinical Implications

Although having small clinical implication, hypothyroidism is associated with significant cardiac function impairment particularly at the subendocardial level, both in systole and diastole. As long as the endocrine pathology is active, the cardiac disturbances are likely to be perpetual, even irreversible. Identifying such patients with impaired cardiac function early on, should guide towards optimum correction of thyroid function and hence saving future irreversible cardiac dysfunction.

### 4.4. Study Limitations

The small number of studied patients and controls reduces the strength of the findings and their clinical application. These remain to be reconfirmed in a larger cohort of patients. A larger group of patients with different hypothyroid severities should assist in identifying clearer relationship with different degrees of LV dysfunction as well as cut-off values that require aggressive treatment. We did not have the possibility of using more advanced techniques for assessing cardiac function, e.g., speckle tracking echocardiography, which might have shed more light on other disturbances.

## 5. Conclusions

In a modest sample of patients with hypothyroidism, in addition to compromised LV diastolic function, LV longitudinal systolic function is also impaired compared to healthy age and gender matched controls. These findings suggest significant subendocardial function impairment, potentially reflecting micro-circulation disease that requires optimum management.

## Figures and Tables

**Table 1 medicina-56-00524-t001:** Clinical and biochemical data in controls vs. hypothyroid patients.

Variable	Controls (*n* = 22)	Hypothyroidism (*n* = 81)	*p* Value
Age (years)	45 ± 14	42 ± 13	0.392
Gender (male, %)	14	8	0.572
Smoking (%)	14	11	0.496
SBP (mmHg)	121 ± 16	123 ± 20	0.236
DBP (mmHg)	82 ± 10	83 ± 16	0.448
Heart rate (beat/min)	76 ± 9	78 ± 15	0.165
Waist (cm)	94 ± 9	91 ± 13	0.452
Hips (cm)	103 ± 9	109 ± 13	0.015
Waist/hip ratio	0.91 ± 0.05	0.84 ± 0.09	<0.001
Weight (kg)	79 ± 14	74 ± 14	0.629
BMI (kg/m^2^)	27 ± 3.1	27 ± 4.9	0.938
BSA (m^2^)	1.1 ± 0.2	1.09 ± 0.2	0.068
Total cholesterol (mmol/L)	5.01 ± 0.9	5.1 ± 1.2	0.651
Triglyceride (mmol/L)	1.58 ± 0.41	1.61 ± 0.85	0.844
Creatinine (μmol/L)	75 ± 15	76 ± 16	0.908
Blood urea nitrogen (mmol/L)	5.4 ± 1.4	5.3 ± 1.9	0.702
Fasting glucose (mmol/L)	5.2 ± 0.7	5.4 ± 0.8	0.476
Red blood cells ×10^6^/L	4.6 ± 0.6	4.3 ± 0.6	0.019
White blood cells ×10^3^/L	7.8 ± 1.3	6.6 ± 2.1	0.011
Hemoglobin (g/dL)	13.4 ± 1.1	12.5 ± 2.9	0.182
Hematocrit (%)	38 ± 8	36 ± 6	0.159
TSH (mU/L)	2.4 ± 1.1	8.3 ± 12.9	0.037

SBP: Systolic blood pressure, DBP: Diastolic blood pressure, BMI: Body mass index, BSA: Body surface area, TSH: Thyroid-stimulating hormone.

**Table 2 medicina-56-00524-t002:** Echocardiographic data in control group vs. hypothyroid patients.

Variable	Controls (*n* = 22)	Hypothyroidism (*n* = 81)	*p* Value
*LV dimension and mass*			
LV mass (g)	169 ± 26	161 ± 32	0.274
LV mass index (g/m ^2.7^)	38 ± 10	42 ± 10	0.078
Inter ventricular septum (cm)	1.0 ± 0.13	1.0 ± 0.14	0.905
LV posterior wall (cm)	0.97 ± 0.1	0.94 ± 0.1	0.245
LV end-diastolic diameter (cm)	4.84 ± 0.4	4.87 ± 0.4	0.706
LV end-systolic diameter (cm)	3.06 ± 0.4	3.13 ± 0.4	0.499
*LV systolic function*			
LV ejection fraction (%)	64 ± 6.9	65 ± 8.5	0.734
LV shortening fraction (%)	34 ± 4.8	36 ± 6.4	0.064
Stroke volume (mL)	75 ± 20	74.7 ± 18	0.876
Lateral s’ (cm)	8.0 ± 1.7	6.5 ± 1.1	<0.001
Septal s’ (cm/s)	8.1 ± 1.6	6.2 ± 1.0	<0.001
Septal MAPSE (cm)	1.53 ± 0.2	1.46 ± 0.2	0.199
Lateral MAPSE (cm)	1.66 ± 0.2	1.49 ± 0.3	0.002
*LV diastolic function*			
E wave (cm/s)	77 ± 7.5	66 ± 10	<0.001
A wave (cm/s)	60 ± 9.7	68 ± 14	0.028
E/A ratio	1.3 ± 0.2	1.0 ± 0.2	<0.001
E wave DT (ms)	158 ± 31	178 ± 33	0.010
Lateral e’ (cm/s)	8.8 ± 3.0	8.5 ± 1.7	0.683
Lateral a’ (cm/s)	9.4 ± 2.3	9.0 ± 2.8	0.571
E/e’ ratio	9.4 ± 2.4	7.8 ± 1.7	<0.001
Septal e’ (cm/s)	8.3 ± 1.5	8.9 ± 2.8	0.327
Septal a’ (cm/s) *LV dyssynchroneus function*	7.4 ± 1.6	8.3 ± 2.0	0.071
IVRT (ms) FT (ms) ET (ms) T-IVT Tei index	93 ± 29 376 ± 91 307 ± 37 8.8 ± 5.1 0.42 ± 0.25	96 ± 21 408 ± 113 297 ± 36 7.5 ± 5.6 0.34 ± 0.24	0.537 0.215 0.524 0.540 0.201
*LA dimensions* LA transversal diameter (cm)	3.6 ± 0.3	3.7 ± 0.4	0.632

LV: left ventricular; LA: left atrium; MAPSE: Mitral annular plane systolic excursion; IVRT: Isovolumic relaxation time; EDT: E wave deceleration time; A: atrial diastolic velocity; a’: late diastolic myocardial velocity; s’: systolic annular velocity, E: early mitral inflow velocity; e’: early annular diastolic velocity; A: late mitral inflow velocity; a’: late annular diastolic velocity; *e*’: early diastolic myocardial velocity; *s*’*:* systolic myocardial velocity; FT: filling time; ET: ejection time; T-IVT: total isovolumic time.

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
