# Peer review of "Left Ventricular Diastolic and Systolic Functions in Patients with Hypothyroidism†"

_medicina, 2020, doi:10.3390/medicina56100524_

Round 1

Reviewer 1 Report

The suggested corrections are ok, except:

1) Incorrect indent in references 21 and 26. 

Reviewer 2 Report

The authors have made the necessary changes to the manuscript and is acceptable.

Thank you and good luck

Author Response

This manuscript is a resubmission of an earlier submission. The following is a list of the peer review reports and author responses from that submission.

Round 1

Reviewer 1 Report

The authors are comparing hypothyroidism patients with controls to determine if the left ventricular diastolic and systolic functions are compromised in those patients. It is a good study, my only concern is that the number of controls is very low as compared to the patients (22 controls and 81 patients have been used). This being said, the authors themselves have noted this fact in their 'study limitation' section. So, in order to forward these findings before going into more cohort of patients, I think it is a good study. 

Minor revision:

Please check grammar throughout the manuscript.

Thank you

Author Response

Comments from Reviewer #1:
The authors are comparing hypothyroidism patients with controls to determine if the left ventricular diastolic
and systolic functions are compromised in those patients. It is a good study, my only concern is that the
number of controls is very low as compared to the patients (22 controls and 81 patients have been used). This
being said, the authors themselves have noted this fact in their 'study limitation' section. So, in order to
forward these findings before going into more cohort of patients, I think it is a good study.
Minor revision:
Please check grammar throughout the manuscript.
Thank you

Response: Thank you for your comments. We tried to improve Grammar in whole paper.

Reviewer 2 Report

Left ventricular diastolic and systolic functions are compromised in patients with hypothyroidism

The authors of this study assess the effects of hypothyroidism on the left ventricular systolic and diastolic functions. The results show disorders in different  echocardiographic values. Since these disorders are persistent, it is necessary to investigate its origin and adjust hypothyroidism treatment.

Corrections

1) Introduction

1.1. “Despite the available evidence on the impairment of cardiovascular system in patients with hypothyroidism, results remain inconsistent and could even be controversial (5– 7)(8–11)”

References 5 to 11 refer to patients with subclinical hypothyroidism but the authors do not specify if their patients have clinical or subclinical hypothyroidism. In case of clinical hypothyroidism, they must add information of studies focused on this type of patients.

1.2. References must be placed in square brackets [ ]. If they are consecutive, they must be together. In the Introduction, the reference “(5-7)(8-11)” must be [5-11].

2) Patients and methods\Patients

The authors do not specify if hypothyroidism is clinical/overt or subclinical or if the patients are under any treatment for hypothyroidism. Controls have normal renal function but nothing is said about the experimental group.

3) Cardiac structure and function\LV structure measurements:

3.1. “…validated formula of Devereux et al. (12) (13) and was indexed to height 2.7 (LVMI) (14)”.

References must be [12,13]. Is the superscript “2,7” a reference?

3.2. There are two numbers in brackets-(1),(13)- next to the formula  “LVM(g)=0.8x(1.04x(LVEDD+PWTD+IVSTD)3-(LVEDD)3)+0.6”, which one is the reference?

4) Cardiac structure and function\LV function measurements:

4.1. “From the spectral wave Doppler LV filling filling”.

Repeated word

4.2. “…septal and lateral e’velocities (16) (17)”.

References must be [16,17]

5) Cardiac structure and function\LV structure measurements:

5.1. “We followed the methods of our previous study, Poniku A, et al. 2018 (21), where are prescribed in detail these measurements.”

Since the citation is the same as in the previous referenced article, this explanation should be at the beginning of the section instead of at the end.

6) Statistical analysis

6.1. “Pearson correlations were performed to identify simple correlations between variables.”

There are no results from Pearson correlations.

6.2. “windows” is written with capital W.

7) Results

7.1. “Age, gender and CV risk factors (smoking, diabetes, arterial hypertension and cholesterol level) were not different between groups”.

In “Patients”, the authors specified that subjects with arterial hypertension were excluded. Maybe, they mean arterial pressure. Correct.

7.2. Table 1:

In the smoking variable, how can they get 7%? 1 person out of 22 is 4,5%, 2 people are 9%. Check data of this table. If these data refer to the number of subjects (n), the name of the variable must be corrected since there is a % sign.

Adding and comparing information about TSH levels could be interesting, if they are available.

7.3. Table 2:

E/e' ratio is also significant (p<0.001). Specify in the text. For example:

“Patients with hypothyroidism had higher A wave velocity (p=0.028), lower E wave velocity, lower E/A ratio (p<0.001, for both), E/e' ratio (p<0.001) and longer E wave DT (p=0.01), compared to controls.”

8) Data interpretation:

8.1. The explanation is confusing and dense, which makes it difficult to understand. It should be rewritten in a clearer way.

8.2. “Many studies have already shown that diastolic dysfunction is hardly an isolated phenomenon but likely reflecting a degree of systolic dysfunction, even at subendocardial level.”

Which studies? The authors do not indicate any study to support that statement.

9) References

References 1, 2 and 3 are incomplete.

Incorrect indent in reference 21.

Check the references.

Author Response

Comments from Reviewer #2:
Left ventricular diastolic and systolic functions are compromised in patients with hypothyroidism
The authors of this study assess the effects of hypothyroidism on the left ventricular systolic and diastolic
functions. The results show disorders in different echocardiographic values. Since these disorders are
persistent, it is necessary to investigate its origin and adjust hypothyroidism treatment.
Corrections
1) Introduction
1.1. “Despite the available evidence on the impairment of cardiovascular system in patients with
hypothyroidism, results remain inconsistent and could even be controversial (5– 7)(8–11)”
References 5 to 11 refer to patients with subclinical hypothyroidism but the authors do not specify if their
patients have clinical or subclinical hypothyroidism. In case of clinical hypothyroidism, they must add
information of studies focused on this type of patients.

Response: We corrected the references. All included patients had clinical hypothyroidism which is now mentioned in the Methods section.

1.2. References must be placed in square brackets [ ]. If they are consecutive, they must be together. In the
Introduction, the reference “(5-7)(8-11)” must be [5-11].

Response: Thank you for your comment. We have corrected the references and brackets.

2) Patients and methods\Patients

Response: Thank you for your comment. We have corrected it.

2) The authors do not specify if hypothyroidism is clinical/overt or subclinical or if the patients are under any
treatment for hypothyroidism. Controls have normal renal function but nothing is said about the experimental
group.

Response:  Thank you for your comment. Now we have added in the Methods section that study patients had clinical hypothyroidism and were under treatment. Also, we have mentioned that patients with impaired renal function were not included in the study.

3) Cardiac structure and function\LV structure measurements:
3.1. “…validated formula of Devereux et al. (12) (13) and was indexed to height 2.7 (LVMI) (14)”.
References must be [12,13]. Is the superscript “2,7” a reference?

Response: We apologize for the error. We have corrected the references. 2.7 is the part of the unit for LVMI (g/m2.7).

3.2. There are two numbers in brackets-(1),(13)- next to the
formula “LVM(g)=0.8x(1.04x(LVEDD+PWTD+IVSTD)3-(LVEDD)3)+0.6”, which one is the reference?

Response:  We have corrected this and deleted (1).

4) Cardiac structure and function\LV function measurements:
4.1. “From the spectral wave Doppler LV filling filling”.
Repeated word

Response:  We apologize for the error. We have corrected it.

4.2. “…septal and lateral e’velocities (16) (17)”.
References must be [16,17]

Response:  We have corrected the references numbers.

5) Cardiac structure and function\LV structure measurements:
5.1. “We followed the methods of our previous study, Poniku A, et al. 2018 (21), where are prescribed in detail
these measurements.”
Since the citation is the same as in the previous referenced article, this explanation should be at the beginning
of the section instead of at the end.

Response:  We agree with your suggestion, but it was the request of the Assistant Editor to put it at the end of the section. 

6) Statistical analysis
6.1. “Pearson correlations were performed to identify simple correlations between variables.”
There are no results from Pearson correlations.

Response:  It was an error in the text. We have removed this sentence.

6.2. “windows” is written with capital W.

Response:  We corrected it.

7) Results
7.1. “Age, gender and CV risk factors (smoking, diabetes, arterial hypertension and cholesterol level) were not
different between groups”.
In “Patients”, the authors specified that subjects with arterial hypertension were excluded. Maybe, they mean
arterial pressure. Correct.

Response:  We have corrected this.

3
7.2. Table 1:
In the smoking variable, how can they get 7%? 1 person out of 22 is 4,5%, 2 people are 9%. Check data of this
table. If these data refer to the number of subjects (n), the name of the variable must be corrected since there is
a % sign. Adding and comparing information about TSH levels could be interesting, if they are available.

Response: We have corrected this. We also added the values in TSH both in patients and controls.

7.3. Table 2:
E/e' ratio is also significant (p<0.001). Specify in the text. For example:
“Patients with hypothyroidism had higher A wave velocity (p=0.028), lower E wave velocity, lower E/A ratio
(p<0.001, for both), E/e' ratio (p<0.001) and longer E wave DT (p=0.01), compared to controls.”

Response: Thank you for your suggestion. We have corrected it.

8) Data interpretation:
8.1. The explanation is confusing and dense, which makes it difficult to understand. It should be rewritten in a
clearer way.

Thank you. We have now clarified the data interpretation.

8.2. “Many studies have already shown that diastolic dysfunction is hardly an isolated phenomenon but likely
reflecting a degree of systolic dysfunction, even at subendocardial level.”
Which studies? The authors do not indicate any study to support that statement.

Response: Thanks for your comment. We added 2 references.

9) References
References 1, 2 and 3 are incomplete.
Incorrect indent in reference 21.
Check the references.

Response: We apologize for the error which has now been corrected.